# EASY-NET Program: Effectiveness of an Audit and Feedback Intervention in the Emergency Care for Acute Conditions in the Lazio Region

**DOI:** 10.3390/healthcare12070733

**Published:** 2024-03-27

**Authors:** Laura Angelici, Carmen Angioletti, Luigi Pinnarelli, Paola Colais, Antonio Giulio de Belvis, Andriy Melnyk, Emanuele La Gatta, Sara Farchi, Marina Davoli, Nera Agabiti, Anna Acampora

**Affiliations:** 1Department of Epidemiology, Regional Health Service–Lazio, Via Cristoforo Colombo, 112, 00147 Rome, Italy; l.pinnarelli@deplazio.it (L.P.); p.colais@deplazio.it (P.C.); m.davoli@deplazio.it (M.D.); n.agabiti@deplazio.it (N.A.); a.acampora@deplazio.it (A.A.); 2Management and Health Laboratory, Institute of Management, Department Embeds, Sant’Anna School of Advanced Studies, Scuola Superiore Sant’Anna, 56127 Pisa, Italy; carmen.angioletti92@gmail.com; 3Critical Pathways and Evaluation Outcome Unit, Fondazione Policlinico Universitario “A. Gemelli”—IRCCS, 00168 Rome, Italy; antonio.debelvis@policlinicogemelli.it; 4Faculty of Economics, Università Cattolica del Sacro Cuore, 00168 Rome, Italy; andriy.melnyk01@icatt.it (A.M.); emanuele.lagatta01@icatt.it (E.L.G.); 5Area Rete Ospedaliera E Specialistica, Direzione Regionale Salute E Integrazione Sociosanitaria Regione Lazio, 00168 Rome, Italy; sfarchi@regione.lazio.it

**Keywords:** acute myocardial infarction, stroke, audit and feedback, emergency networks

## Abstract

The EASY-NET network program (NET-2016-02364191)—effectiveness of audit and feedback (A&F) strategies to improve health practice and equity in various clinical and organizational settings), piloted a novel and more structured A&F strategy. This study compared the effectiveness of the novel strategy against the sole periodic dissemination of indicators in enhancing the appropriateness and timeliness of emergency health interventions for patients diagnosed with acute myocardial infarction (AMI) and ischemic stroke in the Lazio Region. The efficacy of the intervention was assessed through a prospective quasi-experimental design employing a pre- and post-intervention (2021–2022) comparison with a control group. Participating hospitals in the Lazio Region, where professional teams voluntarily engaged in the intervention, constituted the exposed group, while the control group exclusively engaged in routine reporting activities. Effectiveness analysis was conducted at the patient level, utilizing regional health information systems to compute process and outcome indicators. The effectiveness of the intervention was evaluated using difference-in-difference models, comparing pre- and post-intervention periods between exposed and control groups. Estimates were calculated in terms of the difference in percentage points (PP) between absolute risks. Sixteen facilities for the AMI pathway and thirteen for the stroke pathway participated in the intervention. The intervention yielded a reduction in the proportion of 30-day readmissions following hospitalization for ischemic stroke by 0.54 pp in the exposed patients demonstrating a significant difference of −3.80 pp (95% CI: −6.57; −1.03; 5453 patients, 63.7% cases) in the exposed group compared to controls. However, no statistically significant differences attributable to the implemented A&F intervention were observed in other indicators considered. These results represent the first evidence in Italy of the impact of A&F interventions in an emergency setting, utilizing aggregated data from hospitals involved in the Lazio Region’s emergency network.

## 1. Introduction

There is extensive evidence from every country that there is a gap between the healthcare that patients receive and the recommended practice. Specifically, in Italy there is clear evidence of wide variability among health facilities in health processes and outcomes [1,2].

Audit and feedback (A&F) is a proven and widely used methodology for improving the continuous quality of healthcare. It is essentially based on two aspects: audit, a systematic review of the quality of processes and outcomes of care aimed at identifying and measuring critical issues through the definition of criteria, indicators, and standards to compare, and feedback, returning the summary reports of the results of performance evaluation to health professionals involved to promote change [3,4,5,6,7]. 

A&F interventions produce improvements in the professional practice to varying degrees [3,4,5]. A Cochrane review published in 2012 concludes that A&F is effective with an absolute improvement of 4.3% (range interquartile 0.5; 16%) in adherence to evidence-based clinical practice recommendations. The change seems modest in absolute terms, but the cumulative gain resulting from repeated cycles of A&F can lead to large transformations. The ways in which A&F is implemented are widely varied among different studies and contexts, [8,9,10] and, moreover, the scientific progress on these important aspects over the last 20 years has been minimal [11,12,13].

The effectiveness of A&F can be increased if the feedback is posed by a colleague or supervisor, if it is performed more than once, if it is offered in both verbal and written form, and if it includes specific goals to achieve and an action plan to implement the changes. These and other recommendations on how to perform A&F optimally were the subject of a recent paper published by Brehaut et al. [14]. 

The experience of the ASPIRE (Action to Support Practice Implement Research Evidence) project in the UK provided concrete evidence of effectiveness on a high burden disease and applied to larger populations through recommendation packages also based on the A&F tool, as it resulted in the management of chronic pain in primary care [15].

The evidence on how well these recommendations is actually applied in A&F practice is still scarce [16].

In Italy, the utilization of A&F strategies remains limited in certain contexts and is infrequently documented in scientific studies. Remarkably, among the 140 studies scrutinized in the 2012 Cochrane review [11], merely one was conducted in Italy. This stark discrepancy poses significant challenges regarding the transferability of meta-analysis efficacy findings to the Italian context. Consequently, there is a pressing need to conduct experimental studies that delve into both general and context-specific barriers and facilitators.

As part of the EASY-NET project (NET-2016-02364191) [17,18], Work Package 1 (WP1) Lazio Emergency, led by the Department of Epidemiology of the Regional Health Service (RHS)—known as DEP Lazio—conducted a comparative analysis of the effectiveness in enhancing the appropriateness and timeliness of emergency healthcare interventions for acute myocardial infarction (AMI) and ischemic stroke. This comparison was between a structured A&F strategy and the voluntary consultation of numerous process and outcome indicators, updated annually (referred to as the “standard strategy”), facilitated through a dedicated regional web platform named P.Re.Val.E (Programma Regionale Valutazione Esiti—Regional Program for Outcomes and Processes Evaluation) [2].

In the “standard strategy”, feedback is provided to providers through web publications, with no additional initiatives offered by DEP Lazio. Within the WP1 Lazio Emergency project, a structured A&F intervention has been developed, incorporating the latest evidence in the field to optimize these strategies [14,19,20].

In 2021, Lazio reported 7766 hospitalizations for acute myocardial infarction (AMI) and 3249 for ST-elevation myocardial infarction (STEMI) [2]. The number of hospitalizations for AMI and STEMI appear to have been progressively declining over the past decade, aligning with national and international trends [1]. Furthermore, 30-day mortality, an indicator reflecting, at least in part, the quality of patient care provided, seems to have decreased in recent years for both AMI and STEMI cases [21]. Updated analyses for the Lazio Region up to 2021 indicate a reduction from 9.7 percent to 7.6 percent and from 11.1 percent to 8.8 percent, respectively, compared to 2012 [2,22].

The objective of this study is to conduct a quantitative assessment of the effectiveness of an experimental A&F intervention compared to the “standard strategy” in enhancing the appropriateness and timeliness of emergency healthcare interventions for patients with AMI and stroke in the Lazio Region.

## 2. Materials and Methods

### 2.1. Study Design, Participants, and Patients

The quantitative assessment of effectiveness was carried out through a prospective quasi-experimental pre- and post-intervention study with a control group. The pre- and post-intervention periods considered were the years 2021 and 2022, respectively.

The participants in the intervention, commonly referred to as “recipients”, are teams of professionals, including clinical specialists and healthcare managers, engaged in emergency care for patients with AMI or stroke at hospitals in the Lazio Region. The hospitals exposed to the intervention voluntarily participated in the following formal invitations. The control group engaged in standard reporting activities.

The effectiveness analysis was conducted at the patient level. Thus, during the two periods, before and after the intervention, patients admitted for AMI and/or stroke to hospitals participating in the intervention were considered exposed, while patients admitted for AMI and stroke to hospitals not participating in the intervention served as controls (see Figure 1).

The pre- and post-intervention periods were compared between the exposed group and the controls using two process and/or outcome indicators of greatest interest per condition, as listed below:-A 30-day mortality rate after hospital admission in patients with AMI;-Proportion of PTCA (percutaneous transluminal coronary angioplasty) performed in STEMI (ST-elevation myocardial infarction) patients within 90 min of admittance to the hospital emergency room (ER);-In-hospital mortality in patients with ischemic stroke;-Proportion of hospital readmissions within 30 days of discharge in patients with ischemic stroke.

Detailed information regarding these indicators, including calculation formulas, dimensions, rationale, calculation periods, and links to the calculation protocols, are provided in the relevant Appendix A.

### 2.2. A&F Intervention and Control Group

Hospitals engaged in the intervention undertook the following periodic activities over a span of six months:-Arranging regular meetings to update on project activities, as well as to present and discuss the contents of the feedback;-Subsequent to each meeting, the feedback report was disseminated via email in various formats (comprising a comprehensive main document and a hospital-specific PowerPoint presentation) to the designated contact person within the hospital (pertaining to AMI and/or stroke, respectively). Simultaneously, a form was provided to gather information on audit meetings conducted after the feedback;-Issuing formal invitations to plan and execute audit meetings after each feedback session;-Returning the completed form containing details on the characteristics of the conducted audits (such as date, participants, discussion points on indicators, identification of improvement activities, audit minutes, etc.) to the research group.

Further specifics regarding the implementation of the A&F intervention can be found in Angelici et al. [17].

Control groups were provided with web access to the outcomes of the Regional Program for the Evaluation of Outcomes of Health Interventions (P.Re.Val.E.) [2], overseen by DEP. This program annually publishes process and outcome indicators pertaining to various chronic and acute conditions, including AMI and stroke. Through a specific function accessible via the platform, healthcare entities have the option to initiate an audit procedure involving Lazio DEP. Consultation is initiated by professionals, and the lowest level of aggregation available is at the facility level. Additionally, other comparative data available include information from other facilities, previous time periods, and regional-level metrics.

### 2.3. Data Sources

Pseudo-anonymized data retrieved from the health information systems (HIS) of the Lazio Region were utilized to compute the indicators and to gather variables used as adjustment covariates in the analysis.

Specifically, the data were sourced from the Italian Hospital Discharge Registry (HDR), the Healthcare Emergency Information System (HEIS), and the Tax Registry. The HDR information system contains sociodemographic and clinical data systematically recorded during each hospital admission and discharge across facilities within the Lazio Region. This includes primary and secondary diagnoses as well as all procedures performed. Eligibility and exclusion criteria for the selection of the cohort of interest were determined based on the International Classification of Diseases, Ninth Revision, and Clinical Modification (ICD-9-CM) codes (2019). Codes corresponding to each indicator are provided in Appendix A.

An anonymous identification code, generated by the HIS, served as the reference for the record-linkage process, which was conducted using a deterministic methodology. Data from the HDR were linked with information collected through the Health Emergency Information System (HEIS), which routinely gathers sociodemographic and clinical data pertaining to treatments and visits to all Emergency Departments within Lazio hospitals. Additionally, data from the Tax Registry, which includes information on deaths, and the 2011 Census (Lazio Region Longitudinal Study), containing details on patients’ educational qualifications, were incorporated [23].

By integrating data from these different data sources, a comprehensive socio-demographic and health-related profile was established, enabling the tracing of patients’ clinical histories for the five years preceding the relevant admission.

### 2.4. Variables in Analysis

At patient level, demographic, socioeconomic, and clinical variables were analyzed. Demographic data included sex and age categories. Socioeconomic status was approximated using education level, categorized based on the 2011 census or, if unavailable from this source, from the information documented in the HDR. Education was categorized as follows: Bachelor’s degree, lower-middle high school, middle high school, elementary school or none, and not stated.

The integration of different information sources facilitated the tracing of patients’ clinical histories for the five years leading up to the hospitalization incident of interest (referred to as the hospitalization index). Clinical data, including comorbidities and medications, were retrieved from the hospitalization index, admission to the hospital ER index, or from all hospitalizations or ER admissions within the preceding five years. Further details are provided in Appendix A.

Additionally, contextual variables at the hospital level were analyzed, specifically the type of hospital, according to the Lazio Region adult emergency network [24]. This variable comprised the following categories: Emergency Admission Department level I (EADI), Emergency Admission Department level II (EADII), and hospitals with Emergency Room (ER) for AMI [24] and Neurovascular Treatment Unit level I (NTUI), Neurovascular Treatment Unit level II (NTUII), hospitals with a Neurovascular Treatment Team (NVT), and hospitals without a Neurovascular Treatment Team (noNVT) for stroke [25].

### 2.5. Data Management and Statistical Analysis

Analyses were conducted at patient level, with each analyzed indicator defined as a dichotomous outcome variable (Yes/No) (e.g., death or no death within 30 days of hospital admission for AMI).

Patients admitted to hospitals with a volume of activities lower than 50 were excluded from the analyses to consider the relationship between volumes and outcomes [26] and to ensure more reliable results.

The descriptive analyses of demographic, socioeconomic, and clinical characteristics of patients were performed according to their exposed/control status and outcome, both pre- and post-intervention. Chi-square tests were utilized to calculate *p*-values of association.

The effectiveness of the intervention was assessed using difference-in-difference (DID) models [16,17,18,19,20] to compare changes in outcomes from pre- to post-intervention periods between exposed and control groups. These models accounted for changes in secular trends and controlled for measured and unmeasured confounding factors. DID models were implemented through generalized linear models with a binomial probability distribution and identity as the link function. Estimates from DID models were presented as the difference in absolute risks measured in percentage points (PP).

Hospital and patient level characteristics were expected to confound or to modify the relationship between intervention and outcomes and were evaluated as potential confounding factors or effect modifiers.

The univariate association of each demographic, socioeconomic, clinical, and contextual variable with the outcome of interest was tested, and a stepwise procedure was employed to identify the set of covariates entering the final multivariate model. 

All data were analyzed using SAS version 9.4 (SAS Institute, Cary, NC, USA).

## 3. Results

A total of 18 out of 70 (25.7%) hospitals in the Lazio Region participated in the intervention for a total of 29 clinical pathways: 16 were dedicated to AMI and 13 were focused on stroke management. The list of participating hospitals can be found in the Appendix A.

### 3.1. Participating Hospitals

Out of the 70 hospitals surveyed, 31 (44.3%) reported admitting at least one patient with AMI in 2021 and 2022, with activity volumes of 50 admissions or more. Among these, 15 hospitals (48.4%) were exposed to the A&F intervention while 16 (51.6%) were not. Detailed descriptive information for both exposed and control groups is provided in the Appendix A.

A total of 12196 AMI patients were analyzed, with 5986 (49.1%) admissions in 2021 and 6210 (50.9%) in 2022. Of these, 7002 (57.4%) were admitted to exposed hospitals, and 5194 (42.6%) to non-exposed hospitals (Appendix A). Of the 59 hospitals that admitted at least one patient with STEMI during the same period, 20 had activity volumes of at least 50, with 12 (60%) exposed to the intervention and 8 (40%) not exposed. A total of 5084 STEMI patients were included in the analysis, with 2433 (47.8%) admissions in 2021 and 2651 (52.5%) in 2022. Among these, 3272 (64%) were admitted to exposed hospitals and 1812 (36%) to non-exposed hospitals (Appendix A).

Similarly, 18 out of the 70 hospitals (25.7%) reported admitting at least one patient with ischemic stroke during the same period, with activity volumes of 50 admissions or more. Among these, 10 hospitals (66.6%) were exposed to the A&F intervention, while 8 (44.4%) were not. The descriptive details for both exposed and control groups are available in Appendix A. A total of 5949 ischemic stroke patients were included for calculating in-hospital mortality, with 2954 (49.7%) admissions in 2021 and 2995 (50.3%) in 2022. Among these, 3793 (63.8%) were admitted to exposed hospitals and 2156 (36.2%) to non-exposed hospitals (Appendix A). Additionally, 5453 ischemic stroke patients were analyzed for 30-day readmissions following an ischemic stroke, with 2685 (49.2%) admissions in 2021 and 2768 (50.8%) in 2022. Among these, 3471 (63.7%) were admitted to exposed hospitals and 1982 (36.3%) to non-exposed hospitals (Appendix A).

### 3.2. Patient Populations

#### 3.2.1. AMI Patient Cohort

Out of the total 12196 AMI patients admitted to the hospitals analyzed over a span of two years (2021 and 2022), 70% were male, with an average age of 69 years. Detailed demographic information is provided in Appendix A.

No significant differences were observed in terms of demographic and socioeconomic characteristics between exposed and control patients, either in total or when comparing the individual years of 2021 and 2022. However, statistically significant differences were noted in the frequency of certain clinical conditions between exposed and control groups, including arterial hypertension (*p* = 0.007), chronic kidney disease (*p* = 0.006), previous coronary angioplasty (*p* = 0.003), and previous coronary artery bypass grafting (*p* = 0.026). Additionally, there were significant differences in the type of hospital admission (*p* < 0.001) (refer to Appendix A).

In total, 839 (6.9%) patients admitted for AMI died within 30 days of the initial hospital contact, with 51.3% of these deaths occurring in the post-intervention year of 2022 (see Table 1 and Appendix A). The descriptive findings are presented in Table 1. Higher 30-day mortality rates were significantly associated with lower education levels, female gender, and older age groups. Additionally, several clinical conditions showed associations with the outcome (refer to Table 1).

#### 3.2.2. STEMI Patient Cohort

Out of the total 5984 STEMI patients admitted to the hospitals analyzed over the two-year period, 74% were male, with an average age of 66 years. The detailed descriptive information can be found in the Appendix A.

There were no significant differences observed in terms of demographic and socioeconomic characteristics between exposed and control patients, either when considering the total cohort or when analyzing the years 2021 and 2022 separately. However, statistically significant associations were found between the frequency of certain clinical conditions among exposed and control groups, including obesity at indexed admission (*p* = 0.009), previous myocardial infarction (*p* = 0.017), cardiomyopathies at indexed admission (*p* = 0.001), and prior coronary angioplasty (*p* < 0.001). Moreover, significant differences were noted in the type of hospital admission (*p* < 0.001) (refer to Appendix A).

In total (over 2021 and 2022), 3077 (60.5%) patients underwent PTCA within 90 min of admission to the ER, with 52% of these procedures occurring in the post-intervention year of 2022. The descriptive results are presented in Table 2. A higher proportion of STEMI patients undergoing PTCA within 90 min of ER admission was significantly associated with lower education levels, male gender, and the age group of 58–65. Furthermore, several clinical conditions were associated with this outcome (refer to Table 2 and Appendix A).

#### 3.2.3. Ischemic Stroke Patient Cohort

The eligibility criteria for including patients in the stroke cohorts for calculating the two considered indicators differ; hence, each cohort is described separately.

##### Thirty-Day In-Hospital Mortality after First Hospital Admission in Patients with Ischemic Stroke

Out of the total 5949 ischemic stroke patients admitted to the hospitals analyzed over the two-year period, 54% were male, with an average age of 74 years. The detailed descriptive information is provided in the Appendix A.

No significant differences were found in terms of demographic and socioeconomic characteristics between exposed and control patients, either when considering the total cohort or when analyzing the years 2021 and 2022 separately, except for educational qualification (*p* < 0.001). However, statistically significant differences were observed in the frequency of certain clinical conditions between exposed and control groups, including obesity and anemia at indexed admission (*p* = 0.038 and *p* < 0.001), coagulation defects at indexed admission (*p* = 0.001), other forms of ischemic heart disease previously (*p* = 0.002), previous not well-defined forms and complications of heart disease (*p* = 0.004), rheumatic heart disease (*p* = 0.001), and other previous cardiac conditions (*p* = 0.003). Furthermore, the type of emergency stroke network hospital of admission was also associated with exposure status (*p* < 0.001) (refer to Appendix A).

In total (over 2021 and 2022), 432 (7.3%) patients died in hospital within 30 days of first admission, with 46% of these deaths occurring in 2022. The descriptive results are presented in Table 3. Higher 30-day mortality following the first hospital admission was significantly associated with lower education levels, female gender, and older age groups. Additionally, several clinical conditions were associated with this outcome (refer to Table 3 and Appendix A).

##### Proportion of Hospital Readmission within 30 Days of Discharge for Ischemic Stroke

Out of the total 12.196 AMI patients admitted to the hospitals included in the analyses over the two-year period (2021 and 2022), 70% were male, with an average age of 69 years. The detailed descriptive information is provided in the Appendix A.

There were no significant differences observed in terms of demographic and socioeconomic characteristics between exposed and control patients, either when considering the total cohort or when analyzing the years 2021 and 2022 separately, except for educational qualification (*p* < 0.001). However, statistically significant differences were found in the frequency of certain clinical conditions between exposed and control groups, including anemia and coagulation defects at indexed admission (*p* < 0.001), prior myocardial infarction (*p* = 0.034), other forms of ischemic heart disease (*p* = 0.006), heart failure (*p* = 0.019), rheumatic heart disease at indexed admission (*p* = 0.001), and other cardiac conditions (*p* = 0.019). Additionally, the type of emergency stroke network hospital of admission was also associated with exposure status (*p* < 0.001) (refer to Appendix A).

In total (over 2021 and 2022), 392 (7.2%) patients experienced hospital readmission within 30 days of discharge for ischemic stroke, with 54% of these readmissions occurring in the post-intervention year of 2022. The descriptive results are presented in Table 4. A higher proportion of hospital readmissions within 30 days of discharge for ischemic stroke was significantly associated with being in the older age group (*p* = 0.005). Additionally, several clinical conditions were associated with this outcome (refer to Table 4 and Appendix A).

### 3.3. Intervention Effectiveness Evaluation

The unadjusted and adjusted results of DID models applied to compare each indicator between exposed and control patients in the pre- and post-intervention periods are presented in Table 5 and Table 6 and Figure 2.

For the AMI/STEMI pathway, the adjusted analyses of the 30-day mortality following the first hospital contact of AMI patients indicated a reduction of 0.22 pp from 2021 to 2022 in the exposed group and 0.14 PP in the control group, with a DID estimate of −0.08 PP (95% CI −2.80; 2.65, *p*-value = 0.956), demonstrating a non-significant difference in favor of the exposed group.

The proportion of STEMI patients treated with PTCA within 90 min of ER access increased by 1.31 PP from 2021 to 2022 in the exposed group and by 8.63 PP in the control group, resulting in a DID estimate of −7.29 PP (95% CI −12.75; −1.83, *p*-value = 0.009), indicating a significant difference in favor of the control group.

This result contrasts with the expected higher improvement in the exposed group due to the A&F intervention. Consequently, sensitivity analyses were conducted, excluding one facility that, despite receiving periodic feedback, never provided information on conducted audits and failed to adhere to the A&F intervention protocol, either through the proposed method, verbal communication, or meetings. The results of the sensitivity analysis confirm a trend for greater improvement in the control group, although the estimate loses significance: −4.38 (−10.0; 1.23), *p*-value = 0.1259.

For ischemic stroke patients, the adjusted DID model indicated a reduction in in-hospital mortality by 0.74 from 2021 to 2022 in exposed patients and was increased by 0.26 PP in controls with a non-significant difference in favor of the exposed patients of −0.99 PP (95% CI: −2.93; 0.95, *p*-value = 0.315).

Furthermore, 30-day readmissions after hospitalization for ischemic stroke decreased by 0.54 PP from 2021 to 2022 in the exposed group, while there was an increase of 3.25 PP in the control group. This resulted in a significant difference in favor of the exposed group, with a DID estimate of −3.80 PP (95% CI: −6.57; −1.03, *p*-value = 0.007) (refer to Table 6, Figure 2).

## 4. Discussion

This study assessed the effectiveness of an experimental audit and feedback (A&F) intervention compared to the “usual reporting strategy” (P.Re.Val.E) in enhancing the appropriateness and timeliness of emergency health interventions for patients with AMI and ischemic stroke, utilizing process and outcome indicators. While the intervention primarily targeted in-hospital emergency pathways, all organizations within the regional time-dependent network of Lazio [27] were invited to participate.

The findings demonstrate heterogeneity. In terms of AMI/STEMI conditions, the percentage of PTCA performed in STEMI patients within 90 min improved in both exposed and unexposed groups. However, although the improvement was significantly greater in the latter, this difference was not significant in sensitivity analyses conducted after excluding hospitals that did not effectively implement the intervention. No significant differences were observed in 30-day mortality rates following an AMI admission. Conversely, for ischemic stroke, the percentage of patients readmitted within 30 days of discharge showed a significant reduction in the exposed group compared to an increase in the unexposed group. However, there was no significant effect on in-hospital mortality among patients with ischemic stroke. In general, and consistent with the existing literature, changes in mortality may require longer follow-up periods to demonstrate the effectiveness of quality improvement interventions [3].

The heterogeneity observed could be attributed to various factors that represent potential limitations of the study. Firstly, the post-intervention period analyzed was relatively short. The intervention started in February 2022 and continued until September 2023. For the present analyses, only the first 11 months of the post-intervention period (2022) were included. Additionally, participating hospitals conducted their initial audit meetings within the first six months, subsequent improvement actions were implemented after this. Consequently, changes might have begun to manifest during the final five months of the year.

Another factor contributing to the observed heterogeneity, which could constitute a limitation of the study, is the discrepancy between 2021 and 2022 regarding COVID-19 patient care in some hospitals. Unlike in 2022, in 2021, certain hospitals—both exposed and unexposed—allocated entire wards to the care of COVID-19 patients. This disparity strongly influenced internal processes and patient outcomes in a variable manner. Consequently, there may have been improvements in certain indicators in these hospitals between 2021 and 2022, independent of the intervention, due to their reduced exposure to infection control measures. Unfortunately, the lack of information regarding which hospitals and wards were affected by this situation prevented the correction of these data associations.

Another plausible explanation could be that certain hospitals in the non-exposed group might have implemented other quality improvement initiatives independently of the EASY-NET project. Consequently, this could have influenced the analysis. However, due to the unavailability of this information, it was not possible to adjust the analyses for this factor.

Additionally, as a further limitation, it is important to acknowledge that the intervention solely focused on the in-hospital aspect of the pathway, without considering the broader functioning of the entire network. Within this network, the emergency service (ARES118- Regional Health Emergency Company-Rome, Italy) transports patients to the most suitable hospital emergency room based on their clinical severity. This dynamic could potentially affect the comparison between exposed facilities (which include a large proportion of hospitals providing emergency care of higher intensity) and controls (e.g., Appendix A). Nevertheless, to mitigate this limitation, the analyses were adjusted according to hospital type.

Despite the quantitative results not being conclusive and requiring further analysis over a longer follow-up period, it is important to highlight additional valuable findings regarding the positive impact of the intervention on fostering connections and facilitating discussions and benchmarking among professionals from various disciplines (such as cardiology, neurology, emergency care, health management, and epidemiology), as well as across different hospitals and settings. These interactions significantly contributed to enhancing the quality of data collected within participating institutions, thus enriching the overall health information flow. 

In fact, during the periodic meetings, professionals, researchers, and regional representatives engaged in discussions about the indicator results that did not align with their expectations. This discrepancy often stemmed from errors such as incorrect code usage or inaccuracies in recording procedure times during data entry. Consequently, specific audits targeting data quality were initiated [28]. It is important to note that these quality assessments could potentially influence the evaluation of effectiveness, as the follow-up period did not encompass the phase subsequent to the implementation of actions aimed at improving data quality.

Another notable benefit stemming from the intervention was the collective contribution of all participants towards the development of audit support materials, including reports and audit forms, which were collaboratively agreed upon by professionals and regional representatives. These materials have the potential to extend beyond the confines of the research project and be utilized in daily practice. As suggested by recent publications [14,19], involving recipients from the outset of the audit and feedback (A&F) process—including in the selection and definition of indicators, the design of feedback materials, and the determining of the timing of feedback delivery—can enhance engagement. The integration of both verbal and written feedback, such as through report documentation and in-person meetings, provides recipients with opportunities to discuss their results, challenges, and potential solutions periodically, even in informal social settings. This approach fosters peer collaboration and can bolster motivation, thus mitigating the risk of discontinuation.

Another significant benefit stemming from the intervention was the active involvement of all participants in the development of audit support materials, including reports and audit forms, which were collectively agreed upon by professionals and regional stakeholders. These materials hold enduring utility beyond the conclusion of the research project, aligning with recent publications advocating for recipient involvement at every stage of audit and feedback (A&F) implementation [14,19]. Engaging recipients in the design phase of A&F, encompassing indicator selection and definition, as well as feedback material creation and timing, alongside integrating both verbal and written feedback, has been highlighted as crucial [14,19]. Providing opportunities for periodic discussions in informal settings to review results, address challenges, and foster peer collaboration has been shown to bolster motivation and mitigate discontinuation risks. Notably, all but one hospital participated enthusiastically in all scheduled activities, actively contributing to meeting discussions. Moreover, hospitals were empowered to independently organize periodic audit meetings, thereby enabling the customization of activities to suit the specific contexts of hospitals that varied in size, complexity, volume of activities, and organizational processes. This decentralized approach fosters adaptability and ensures that audit processes remain tailored to the unique needs of each healthcare setting.

In an era marked by increasing complexity and a growing emphasis on value-driven healthcare, the insights derived from this study are poised to enhance evidence-based practice and contribute to the ongoing evolution of healthcare delivery models. Through the provision of feedback and the establishment of cyclical audit processes, this study aims to bolster the effective implementation of networks designed to enhance the appropriateness and timeliness of emergency health interventions, particularly for patients with time-sensitive conditions.

The proposed intervention involved all actors of the time-dependent emergency network in the context of the Lazio Region, although the focus of the activities was the in-hospital pathway. There is evidence that clinical networks can improve the delivery of healthcare, although there are few high-quality quantitative studies of their effectiveness [29]. Organizations in such networks need to collaborate and coordinate their actions to achieve their common purpose. They also need to align goals, balance power, manage conflict, monitor performance, and hold members accountable for network-level outcomes [30].

Although current performance measurement systems in Italy provide feedback on time-dependent care conditions [1,2,31,32], they often fall short in evaluating interventions at the network level. Therefore, there is a pressing need for further research endeavors to adopt a network-centric approach. Such initiatives would enable the comprehensive evaluation and improvement of the entire emergency care pathway, thereby exerting a significant influence on critical junctures across patients’ healthcare journeys. This holistic perspective is crucial for optimizing healthcare delivery and enhancing patient outcomes.

## 5. Conclusions

The findings show, for the first time in Italy, the effects of A&F interventions within an emergency setting, leveraging aggregated data from hospitals participating in the Lazio Region emergency network. The delivery of feedback and the implementation of cyclical audit processes have the potential to facilitate the efficient establishment of networks aimed at enhancing the appropriateness and promptness of emergency healthcare interventions for patients with time-dependent conditions.

## Figures and Tables

**Figure 1 healthcare-12-00733-f001:**
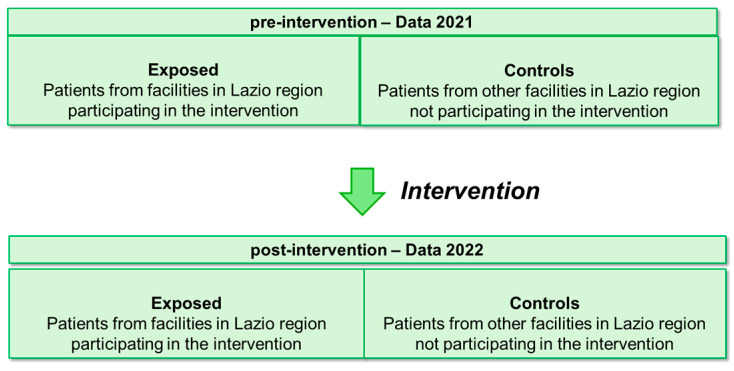
Summary diagram of pre–post intervention design with control group.

**Figure 2 healthcare-12-00733-f002:**
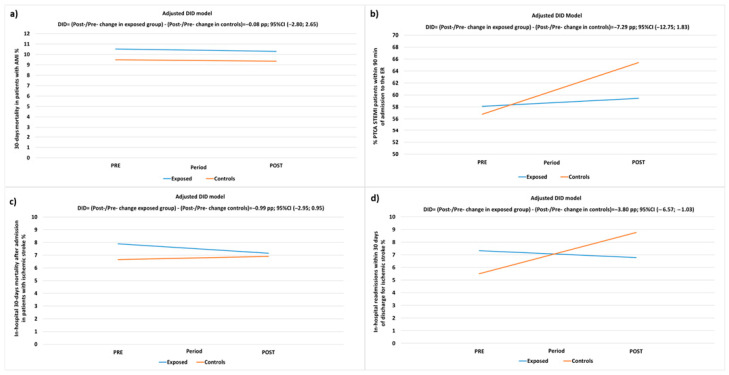
DID-adjusted model for the evaluation of pre-/post-intervention (PP) changes in: (**a**) 30-day mortality after first hospital admission of AMI patients; (**b**) proportion of PTCA performed in STEMI patients within 90 min of admission to the ER; (**c**) in-hospital mortality within 30 days of first hospital admission in patients with ischemic stroke; (**d**) proportion of hospital readmissions within 30 days of discharge for ischemic stroke in exposed patients and controls.

**Table 1 healthcare-12-00733-t001:** Characteristics of patients included in the AMI cohort in 2021 and 2022 from participating facilities by 30-day mortality after first hospital admission (Yes/No).

	Thirty-Day Mortality after First Hospital Admission in Patients with AMI					χ^2^ *p*-Value
	Yes		No		Total	
	N	Row %	N	Row %	N	
Total	839		11,357		12,196	
A&F intervention						0.011
Exposed Groups	517	7.4	6485	92.6	7002	
Control Groups	322	6.2	4872	93.8	5194	
Year						0.842
2021	409	6.8	5577	93.2	5986	
2022	430	6.9	5780	93.1	6210	
Sex						<0.0001
Female	354	9.8	3251	90.2	3605	
Male	485	5.6	8106	94.4	8591	
Age (years)						<0.0001
19–59	60	1.9	3033	98.1	3093	
60–69	92	3.0	2976	97.0	3068	
70–79	220	7.3	2795	92.7	3015	
80–100	467	15.5	2553	84.5	3020	
Education level						<0.0001
Degree	69	5.5	1175	94.5	1244	
Lower-middle high school	215	5.4	3731	94.6	3946	
Middle high school	176	5.1	3292	94.9	3468	
None or elementary school	340	11.2	2692	88.8	3032	
Not stated	39	7.7	467	92.3	506	
Concomitant clinical conditions						
Cancer	97	14.4	577	85.6	674	<0.0001
Diabetes	89	12.3	633	87.7	722	<0.0001
Lipid metabolism disorders	25	7.1	329	92.9	354	0.890
Obesity	5	7.6	61	92.4	66	0.823
Obesity at indexed admission	13	2.4	525	97.6	538	<0.0001
Anemia	62	19.0	264	81.0	326	<0.0001
Anemia at indexed admission	54	9.7	504	90.3	558	0.008
Coagulation defects	1	16.7	5	83.3	6	0.343
Coagulation defects at indexed admission	_	_	3	100.0	3	0.638
Other hematological diseases	2	6.5	29	93.5	31	0.925
Other hematological diseases at indexed admission	4	8.7	42	91.3	46	0.626
Arterial hypertension	112	9.1	1125	90.9	1237	0.001
Previous myocardial infarction	54	5.7	895	94.3	949	0.132
Other forms of ischemic heart disease	90	9.0	905	91.0	995	0.005
Heart failure	90	16.6	452	83.4	542	<0.0001
Not well-defined forms and complications of heart disease	6	8.0	69	92.0	75	0.701
Rheumatic heart disease	6	14.0	37	86.0	43	0.066
Rheumatic heart disease at indexed admission	27	16.9	133	83.1	160	<0.0001
Cardiomyopathies	8	14.5	47	85.5	55	0.024
Cardiomyopathies at indexed admission	8	5.8	131	94.2	139	0.599
Acute endocarditis and myocarditis	_	_	6	100.0	6	0.506
Other cardiac conditions	14	13.3	91	86.7	105	0.009
Other cardiac conditions at indexed admission	35	10.8	289	89.2	324	0.005
Conduction disorders and arrhythmias	75	13.1	499	86.9	574	<0.0001
Cerebrovascular diseases	47	11.4	366	88.6	413	0.000
Cerebrovascular diseases at indexed admission	34	10.2	300	89.8	334	0.016
Vascular diseases	40	13.3	260	86.7	300	<0.0001
Vascular diseases at indexed admission	24	6.5	348	93.5	372	0.741
Chronic obstructive pulmonary disease (COPD)	38	13.2	249	86.8	287	<0.0001
Chronic nephropathy	68	15.7	364	84.3	432	<0.0001
Chronic kidney disease	116	10.9	945	89.1	1061	<0.0001
Chronic diseases (liver, pancreas, intestines)	9	13.2	59	86.8	68	0.038
Chronic diseases (liver, pancreas, intestines) at indexed admission	7	10.0	63	90.0	70	0.301
Previous coronary artery bypass grafting	31	9.7	287	90.3	318	0.041
Previous coronary angioplasty	65	5.3	1168	94.7	1233	0.019
Cerebrovascular revascularization	6	7.9	70	92.1	76	0.726
Other heart surgery	13	17.6	61	82.4	74	0.000
Other vessel surgery	30	11.4	233	88.6	263	0.003
Type of hospital *						0.120
EADI	554	6.8	7575	93.2	8129	
EADII	262	7.3	3327	92.7	3589	
ER	23	4.8	455	95.2	478	

* EADI: Emergency Admission Department level I; EADII: Emergency Admission Department level II; ER: Hospital Emergency Room.

**Table 2 healthcare-12-00733-t002:** Characteristics of patients included in the STEMI cohort in 2021 and 2022 from participating facilities according to performing of PTCA within 90 min of admission to the ER (Yes/No).

	% PTCA STEMI Patients within 90 min of Admission to the ER					χ^2^ *p*-Value
	Yes		No		Total	
	N	Row %	N	Row %	N	
Total	3077		2007		5084	
A&F intervention						0.069
Exposed Groups	1950	59.6	1322	40.4	3272	
Control Groups	1127	62.2	685	37.8	1812	
Year						0.003
2021	1420	58.4	1013	41.6	2433	
2022	1657	62.5	994	37.5	2651	
Sex						<0.0001
Female	701	54.5	586	45.5	1287	
Male	2376	62.6	1421	37.4	3797	
Age (years)						<0.0001
21–57	879	67.3	428	32.7	1307	
58–65	808	66.5	407	33.5	1215	
66–75	781	60.1	519	39.9	1300	
76–100	609	48.3	653	51.7	1262	
Education level						<0.0001
Degree	339	58.8	238	41.2	577	
Lower-middle high school	1031	62.2	626	37.8	1657	
Middle high school	988	63.0	581	37.0	1569	
None or elementary school	548	54.0	467	46.0	1015	
Not stated	171	64.3	95	35.7	266	
Concomitant clinical conditions						
Cancer	109	44.9	134	55.1	243	<0.0001
Diabetes	67	41.6	94	58.4	161	<0.0001
Lipid metabolism disorders	36	45.0	44	55.0	80	0.004
Obesity	7	43.8	9	56.3	16	0.169
Obesity at indexed admission	126	56.8	96	43.2	222	0.240
Anemia	16	26.7	44	73.3	60	<0.0001
Anemia at indexed admission	60	37.7	99	62.3	159	<0.0001
Coagulation defects	1	100.0	_	_	1	0.419
Coagulation defects at indexed admission	1	100.0	_	_	1	0.419
Other hematological diseases	5	33.3	10	66.7	15	0.031
Other hematological diseases at indexed admission	11	64.7	6	35.3	17	0.724
Arterial hypertension	166	51.1	159	48.9	325	0.000
Previous myocardial infarction	81	38.4	130	61.6	211	<0.0001
Other forms of ischemic heart disease	96	44.2	121	55.8	217	<0.0001
Heart failure	27	32.5	56	67.5	83	<0.0001
Not well-defined forms and complications of heart disease	4	30.8	9	69.2	13	0.028
Rheumatic heart disease	4	50.0	4	50.0	8	0.542
Rheumatic heart disease at indexed admission	18	41.9	25	58.1	43	0.012
Cardiomyopathies	3	18.8	13	81.3	16	0.001
Cardiomyopathies at indexed admission	23	39.7	35	60.3	58	0.001
Acute endocarditis and myocarditis	_	_	1	100.0	1	0.216
Other cardiac conditions	2	15.4	11	84.6	13	0.001
Other cardiac conditions at indexed admission	21	31.8	45	68.2	66	<0.0001
Conduction disorders and arrhythmias	45	36.9	77	63.1	122	<0.0001
Cerebrovascular diseases	54	45.0	66	55.0	120	0.000
Cerebrovascular diseases at indexed admission	42	34.4	80	65.6	122	<0.0001
Vascular diseases	28	35.0	52	65.0	80	<0.0001
Vascular diseases at indexed admission	42	39.6	64	60.4	106	<0.0001
Chronic obstructive pulmonary disease (COPD)	23	38.3	37	61.7	60	0.000
Chronic nephropathy	37	38.1	60	61.9	97	<0.0001
Chronic kidney disease	131	44.3	165	55.7	296	<0.0001
Chronic diseases (liver, pancreas, intestines)	9	47.4	10	52.6	19	0.240
Chronic diseases (liver, pancreas, intestines) at indexed admission	13	48.1	14	51.9	27	0.187
Previous coronary artery bypass grafting	19	41.3	27	58.7	46	0.007
Previous coronary angioplasty	159	50.6	155	49.4	314	0.000
Cerebrovascular revascularization	10	62.5	6	37.5	16	0.871
Other heart surgery	6	28.6	15	71.4	21	0.003
Other vessel surgery	25	37.9	41	62.1	66	0.000
Type of hospital *						0.142
EADI	1909	61.3	1204	38.7	3113	
EADII	1168	59.3	803	40.7	1971	

* EADI: Emergency Admission Department level I; EADII: Emergency Admission Department level II.

**Table 3 healthcare-12-00733-t003:** Characteristics of patients included in the ischemic stroke cohort in 2021 and 2022 from participating facilities by in-hospital mortality within 30 days of first hospital admission in patients with ischemic stroke (Yes/No).

	Thirty-Day In-Hospital Mortality after Admission in Patients with Ischemic Stroke					χ^2^ *p*-Value
	Yes		No		Total	
	N	Row %	N	Row %	N	
Total	432		5517		5949	
A&F intervention						0.006
Exposed Groups	249	8.2	2803	91.8	3052	
Control Groups	183	6.3	2714	93.7	2897	
Year						0.081
2021	232	7.9	2722	92.1	2954	
2022	200	6.7	2795	93.3	2995	
Sex						<0.0001
Female	243	8.8	2512	91.2	2755	
Male	189	5.9	3005	94.1	3194	
Age (years)						<0.0001
35–66	26	1.7	1482	98.3	1508	
67–76	71	4.8	1401	95.2	1472	
77–83	111	7.8	1318	92.2	1429	
84–100	224	14.5	1316	85.5	1540	
Education level						<0.0001
Degree	24	4.2	543	95.8	567	
Lower-middle high school	107	6.7	1492	93.3	1599	
Middle high school	76	5.5	1304	94.5	1380	
None or elementary	211	10.0	1901	90.0	2112	
Not stated	14	4.8	277	95.2	291	
Concomitant clinical conditions						
Cancer	21	9.9	192	90.1	213	0.137
Diabetes	34	11.8	254	88.2	288	0.002
Lipid metabolism disorders	7	6.4	103	93.6	110	0.714
Obesity	4	11.1	32	88.9	36	0.372
Obesity at indexed admission	9	6.5	130	93.5	139	0.718
Anemia	23	13.5	147	86.5	170	0.001
Anemia at indexed admission	11	6.1	169	93.9	180	0.546
Coagulation defects	_	_	3	100.0	3	0.628
Coagulation defects at indexed admission	1	8.3	11	91.7	12	0.886
Other hematological diseases	1	6.7	14	93.3	15	0.929
Other hematological diseases at indexed admission	4	9.3	39	90.7	43	0.605
Arterial hypertension	71	12.0	520	88.0	591	<0.0001
Previous myocardial infarction	14	11.1	112	88.9	126	0.092
Other forms of ischemic heart disease	27	10.7	226	89.3	253	0.033
Heart failure	50	17.4	237	82.6	287	<0.0001
Not well-defined forms and complications of heart disease	4	9.8	37	90.2	41	0.537
Rheumatic heart disease	8	17.8	37	82.2	45	0.006
Rheumatic heart disease at indexed admission	2	3.3	58	96.7	60	0.239
Cardiomyopathies	3	10.0	27	90.0	30	0.562
Cardiomyopathies at indexed admission	3	8.3	33	91.7	36	0.804
Acute endocarditis and myocarditis	_	_	3	100.0	3	0.628
Other cardiac conditions	12	14.6	70	85.4	82	0.010
Other cardiac conditions at indexed admission	8	3.5	220	96.5	228	0.026
Conduction disorders and arrhythmias	55	14.6	321	85.4	376	<0.0001
Cerebrovascular diseases	40	9.6	378	90.4	418	0.059
Vascular diseases	11	8.0	126	92.0	137	0.726
Vascular diseases at indexed admission	15	6.0	233	94.0	248	0.452
Chronic obstructive pulmonary disease (COPD)	21	16.9	103	83.1	124	<0.0001
Chronic nephropathy	26	16.4	133	83.6	159	<0.0001
Chronic nephropathy at indexed admission	29	13.2	191	86.8	220	0.001
Chronic kidney disease	3	8.6	32	91.4	35	0.765
Diseases chronic diseases (liver, pancreas, intestines)	2	4.8	40	95.2	42	0.531
Diseases chronic diseases (liver, pancreas, intestines) at indexed admission	9	6.3	135	93.8	144	0.636
Cerebrovascular revascularization	_	_	27	100.0	27	0.145
Other heart surgery	13	19.4	54	80.6	67	<0.0001
Other vessel surgery	11	8.5	119	91.5	130	0.594
Type of hospital *						0.207
noNVT	15	12.0	110	88.0	125	
NVT	11	6.3	164	93.7	175	
NTUI	209	7.1	2742	92.9	2951	
NTUII	197	7.3	2501	92.7	2698	

* noNVT: Hospital without Neurovascular Treatment Team; NVT: Hospital with a Neurovascular Treatment Team; NTUI: Neurovascular Treatment Unit level I; NTUII: Neurovascular Treatment Unit level II.

**Table 4 healthcare-12-00733-t004:** Characteristics of patients included in the stroke cohort in 2021 and 2022 from participating facilities according to hospital readmissions within 30 days of discharge for ischemic stroke (Yes/No).

	% of Hospital Readmissions within 30 days of Discharge for Ischemic Stroke					χ^2^ *p*-Value
	Yes		No		Total	
	N	Row %	N	Row %	N	
Total	392		5061		5453	
A&F intervention						0.910
Exposed Groups	201	7.8	2580	92.8	2781	
Control Groups	191	7.7	2481	92.9	2672	
Year						0.248
2021	182	7.3	2503	93.2	2685	
2022	210	8.2	2558	92.4	2768	
Sex						0.062
Female	159	6.9	2321	93.6	2480	
Male	233	8.5	2740	92.2	2973	
Age (years)						0.005
35–66	71	5.5	1296	94.8	1367	
67–76	101	8.1	1240	92.5	1341	
77–83	125	9.4	1323	91.4	1448	
84–100	95	7.9	1202	92.7	1297	
Education level						0.432
Degree	41	8.2	498	92.4	539	
Lower-middle high school	96	6.9	1384	93.5	1480	
Middle high school	90	7.5	1202	93.0	1292	
None or elementary school	149	8.7	1719	92.0	1868	
Not stated	16	6.2	258	94.2	274	
Concomitant clinical conditions						
Cancer	16	9.1	175	91.6	191	0.518
Diabetes	18	7.8	232	92.8	250	0.994
Lipid metabolism disorders	7	7.5	93	93.0	100	0.941
Obesity	4	14.3	28	87.5	32	0.243
Obesity at indexed admission	4	3.2	125	96.9	129	0.069
Anemia	12	9.0	133	91.7	145	0.608
Anemia at indexed admission	12	8.1	149	92.5	161	0.895
Coagulation defects	1	50.0	2	66.7	3	0.080
Coagulation defects at indexed admission	1	11.1	9	90.0	10	0.731
Other hematological diseases	1	7.7	13	92.9	14	0.995
Other hematological diseases at indexed admission	2	5.6	36	94.7	38	0.645
Arterial hypertension	35	7.3	481	93.2	516	0.708
Previous myocardial infarction	7	6.7	104	93.7	111	0.716
Other forms of ischemic heart disease	17	8.3	205	92.3	222	0.782
Heart failure	27	13.0	207	88.5	234	0.009
Not well-defined forms and complications of heart disease	4	12.1	33	89.2	37	0.392
Rheumatic heart disease	5	16.1	31	86.1	36	0.118
Rheumatic heart disease at indexed admission	10	21.3	47	82.5	57	0.002
Cardiomyopathies	1	3.8	26	96.3	27	0.482
Cardiomyopathies at indexed admission	3	10.0	30	90.9	33	0.671
Acute endocarditis and myocarditis	_	_	3	100.0	3	0.630
Other cardiac conditions	8	13.3	60	88.2	68	0.142
Other cardiac conditions at indexed admission	16	7.9	203	92.7	219	0.945
Conduction disorders and arrhythmias	30	10.5	286	90.5	316	0.102
Cerebrovascular diseases	32	9.4	340	91.4	372	0.274
Vascular diseases	11	9.6	114	91.2	125	0.481
Vascular diseases at indexed admission	11	5.0	221	95.3	232	0.140
Chronic obstructive pulmonary disease (COPD)	10	10.8	93	90.3	103	0.318
Chronic nephropathy	18	15.9	113	86.3	131	0.003
Chronic nephropathy at indexed admission	23	13.9	166	87.8	189	0.007
Chronic kidney disease	6	23.1	26	81.3	32	0.011
Diseases chronic diseases (liver, pancreas, intestines)	3	8.3	36	92.3	39	0.903
Diseases chronic diseases (liver, pancreas, intestines) at indexed admission	9	7.3	123	93.2	132	0.868
Cerebrovascular revascularization	_	_	27	100.0	27	0.147
Other heart surgery	7	15.2	46	86.8	53	0.088
Other vessel surgery	11	10.5	105	90.5	116	0.334
Type of hospital *						0.551
noNVT	8	7.8	102	92.7	110	
NVT	16	11.0	146	90.1	162	
NTUI	197	7.9	2506	92.7	2703	
NTUII	171	7.4	2307	93.1	2478	

* noNVT: Hospital without Neurovascular Treatment Team; NVT: Hospital with a Neurovascular Treatment Team; NTUI: Neurovascular Treatment Unit level I; NTUII: Neurovascular Treatment Unit level II.

**Table 5 healthcare-12-00733-t005:** Unadjusted and adjusted DID model results for the assessment of change (PP; 95% CI) pre-/post-intervention for indicators in AMI/STEMI pathway.

AMI/STEMI Pathway						
Indicators	Facilities (Patients)	30-Day Mortality after First Hospital Admission in Patients with AMI		Facilities (Patients)	% PTCA-STEMI Patients within 90 min of Admission to the ER	
		Unadjusted	Adjusted *		Unadjusted	Adjusted **
Exposed Group						
“Pre-”—2021 (%)	12 (3393)	7.40	10.51	12 (1541)	58.66	58.12
“Post-”—2022 (%)	15 (3609)	7.37	10.29	12 (1731)	60.43	59.43
Difference Post-/Pre- (PP)		−0.03	−0.22		1.77	1.31
Control Group						
“Pre-”—2021 (%)	15 (2593)	6.09	9.48	8 (892)	57.85	56.79
“Post”—2022 (%)	16 (2601)	6.31	9.34	8 (920)	66.41	65.42
Difference Post-/Pre- (PP)		0.22	−0.14		8.56	8.63
DID PP		−0.24	−0.08		−6.80	−7.29
(95% CI)		(−2.03; 1.56)	(−2.80; 2.65)		(−12.38; −1.22)	(−12.75; −1.83)
*p*-value for interaction		0.794	0.956		0.017	0.009

DID: Post-/Pre-intervention changes in the exposed groups and post-/pre-intervention changes in controls in percentage points (PP). After a stepwise procedure on all variables associated with the outcome: * the model was adjusted for sex, age in classes, cancer, anemia, prior myocardial infarction, heart failure, diabetes, and prior systolic coronary angioplasty; ** the model was adjusted for sex, age in classes, cancer, anemia present at the STEMI episode, prior myocardial infarction, prior conduction disorders, prior arrhythmias, and chronic nephropathy present at the STEMI episode.

**Table 6 healthcare-12-00733-t006:** Unadjusted and adjusted DID model results for assessment of change (PP; 95% CI) pre/post-intervention for indicators in ischemic stroke pathway.

Ischemic Stroke Pathway						
Indicators	Facilities (Patients)	Thirty-Day In-Hospital Mortality after Admission in Patients with Ischemic Stroke		Facilities (Patients)	% of Hospital Readmissions within 30 days of Discharge for Ischemic Stroke	
		Unadjusted	Adjusted *		Unadjusted	Adjusted **
Exposed Group						
“Pre-”—2021 (%)	8 (1885)	8.54	7.89	8 (1711)	7.36	7.33
“Post-”—2022 (%)	8 (1908)	6.87	7.15	8 (1760)	6.88	6.79
Difference Post-/Pre- (PP)		−1.67	−0.74		−0.48	−0.54
Control Group						
“Pre-”—2021 (%)	8 (1069)	6.64	6.65	8 (974)	5.75	5.51
“Post-”—2022 (%)	9 (1087)	6.35	6.91	9 (1008)	8.83	8.76
Difference Post-/Pre- (PP)		−0.29	0.26		3.08	3.25
DID PP		−1.38	−0.99		−3.57	−3.80
(95% CI)		(−4.07; 1.30)	(−2.93; 0.95)		(−6.42; −0.72)	(−6.57; −1.03)
*p*-value for interaction		0.313	0.315		0.014	0.007

DID: Changes in exposed groups post-/pre-intervention and changes in controls post-/pre-intervention in percentage points (PP). After a stepwise procedure on all variables associated with the outcome: * the model was adjusted for gender, age in classes, chronic nephropathy, and rheumatic heart disease at admission for ischemic stroke; ** the model was adjusted for gender, age in classes, heart failure, cerebrovascular disease at the time of admission for ischemic stroke, and hypertension.

## Data Availability

Data related to the findings reported in our manuscript are available to all interested researchers upon reasonable request and with the permission of the Regional Department because of stringent legal restrictions regarding the privacy policy on personal information in Italy (national legislative decree on privacy policy no. 196/30 June 2003). For these reasons our dataset cannot be made available in a public data deposition.

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
