# Peer review of "EASY-NET Program: Effectiveness of an Audit and Feedback Intervention in the Emergency Care for Acute Conditions in the Lazio Region"

_healthcare, 2024, doi:10.3390/healthcare12070733_

Round 1

Reviewer 1 Report

Comments and Suggestions for Authors

Thank you for possibility to review  'EASY-NET Program: effectiveness of an Audit and Feedback Intervention in the Emergency Care for acute conditions in Lazio region'.  The publication draws attention to the need of increased quality control in healthcare. In my opinion the paper may be accepted after minor revisions. 

Comment 1: 

Please consider revision of the manuscript by native speaker. The English language is not clear in some parts

Comment 2:

The amount of data contained in the paper is very large. Wouldn't division into parts be more accessible to the reader?

Comment 3:

 In line 95 and 99 IMA abbreviation was used. Is this a typo?

Comment 4:

Line 350 is incomprehensible to me.

In my opinion the article deserves publication in your respected journal after minor revision and would be valuable to readers working in the field of epidemiology and quality management. In order to expand the group of readers it would be necessary to simplify it.

Yours sincerely,

Reviewer 2 Report

Comments and Suggestions for Authors

I have reviewed the manuscript titled "EASY-NET Program: Effectiveness of an Audit and Feedback Intervention in Emergency Care for Acute Conditions in the Lazio Region," which aims to perform a quantitative evaluation of the effectiveness of an experimental A&F intervention compared to the standard strategy. While the study addresses an important topic through a quasi-experimental approach, it presents various factors that need careful consideration.

My primary concern with this manuscript is the ethical aspect of the research. The authors mention that the study was not evaluated by an Institutional Review Board (IRB) and provide reasons for this decision. However, according to the Declaration of Helsinki, an ethics and research committee should review and approve any research involving human subjects. Additionally, they state that informed consent does not apply, which is also concerning, as there seem to be no justifiable reasons for exempting participants or their families from informed consent.

Other aspects of the manuscript that draw attention include the citation starting with number 3, leaving the whereabouts of the first two citations unclear.

The authors do not address a limitations section in the discussion.

Are all 27 supplementary tables or figures essential? An abundance of information may divert readers from the study's main objective. In the same context, the Supplementary Materials section mentions only 25 of the 27 supplementary resources.

Comments on the Quality of English Language

No significant grammatical errors were identified in the manuscript.

Round 2

Reviewer 2 Report

Comments and Suggestions for Authors

The authors have highlighted the ethical implications of their study; likewise, they made various adjustments requested by the evaluator.

Comments on the Quality of English Language

No comments regarding the language.